# New Approach Studying Interactions Regarding Trade-Off between Beef Performances and Meat Qualities

**DOI:** 10.3390/foods8060197

**Published:** 2019-06-07

**Authors:** Alexandre Conanec, Brigitte Picard, Denis Durand, Gonzalo Cantalapiedra-Hijar, Marie Chavent, Christophe Denoyelle, Dominique Gruffat, Jérôme Normand, Jérôme Saracco, Marie-Pierre Ellies-Oury

**Affiliations:** 1Universite Clermont Auvergne, INRA, VetAgro Sup, UMR Herbivores, F-63122 Saint-Genes-Champanelle, France; alexandre.conanec@agro-bordeaux.fr (A.C.); brigitte.picard@inra.fr (B.P.); denis.durand@inra.fr (D.D.); gonzalo.cantalapiedra@inra.fr (G.C.-H.); dominique.gruffat@inra.fr (D.G.); 2Contrôle de Qualité et Fiabilité Dynamique (CQFD) team, Inria BSO, F-33400 Talence, France; marie.chavent@math.u-bordeaux.fr (M.C.); jerome.saracco@math.u-bordeaux.fr (J.S.); 3Universite de Bordeaux, IMB, UMR 5251, F-33400 Talence, France; 4Institut de l’Elevage, Service Qualite des Carcasses et des Viandes, 69007 Lyon, France; christophe.denoyelle@idele.fr (C.D.); jerome.normand@idele.fr (J.N.); 5ENSC Bordeaux INP, IMB, UMR 5251, F-33400 Talence, France

**Keywords:** trade-off, meat quality, beef performances, modeling

## Abstract

The beef cattle industry is facing multiple problems, from the unequal distribution of added value to the poor matching of its product with fast-changing demand. Therefore, the aim of this study was to examine the interactions between the main variables, evaluating the nutritional and organoleptic properties of meat and cattle performances, including carcass properties, to assess a new method of managing the trade-off between these four performance goals. For this purpose, each variable evaluating the parameters of interest has been statistically modeled and based on data collected on 30 Blonde d’Aquitaine heifers. The variables were obtained after a statistical pre-treatment (clustering of variables) to reduce the redundancy of the 62 initial variables. The sensitivity analysis evaluated the importance of each independent variable in the models, and a graphical approach completed the analysis of the relationships between the variables. Then, the models were used to generate virtual animals and study the relationships between the nutritional and organoleptic quality. No apparent link between the nutritional and organoleptic properties of meat (*r* = −0.17) was established, indicating that no important trade-off between these two qualities was needed. The 30 best and worst profiles were selected based on nutritional and organoleptic expectations set by a group of experts from the INRA (French National Institute for Agricultural Research) and Institut de l’Elevage (French Livestock Institute). The comparison between the two extreme profiles showed that heavier and fatter carcasses led to low nutritional and organoleptic quality.

## 1. Introduction

The meat sector is facing various challenges in France, due to a current context of high quality demand and competiveness. Indeed, consumers expect homogeneous organoleptic quality, and more recently, consumers’ expectations in terms of healthiness and the nutritional quality of food are increasing. The beef sector is obviously concerned, due to the lower meat consumption in the past few years [1]. In addition, farmers and retailers are facing low margins in a competitive sector. Therefore, to preserve the beef cattle industry in the territory, the sustainability of these main actors has to be enhanced by rearing more efficient animals to decrease the cost of production and provide standardized carcasses of high quality. Despite the stakes, few studies have been carried out to tackle the problem as a whole, describing the relation between these four parameters of interest: animal performances, carcass properties, nutritional qualities, and organoleptic qualities. Among these studies, a creative approach proceeding to a double dimension reduction (via a clustering of variables followed by a principal component analysis) on a dataset made of multiple variables measuring carcass value, fatty acid profile, and sensory tasting descriptors concluded that there is no relationship between the nutritional and sensory quality of the meat produced by young bulls [2]. Furthermore, other studies have highlighted some correlations (positive and negative) between these two qualities. For instance, it has been reported that a high amount of intramuscular fat enhances the flavor, the juiciness, and the tenderness of the meat [3,4], but reduces the proportion of polyunsaturated fatty acid (PUFA), whereas saturated (SFA) and monounsaturated fatty acids (MUFA) proportions increase [5]. The first objective of the present work was to integrate these four parameters of interest to analyze the relationships between them. A second aim was to set up a method of managing the trade-off to help the beef cattle industry design specifications that take the possible interaction between the nutritional and organoleptic qualities of the meat into account.

## 2. Materials and Methods 

The whole method is based on a dataset of 30 animals, which will be described further. First, 62 variables measured on those animals were pre-treated to reduce the redundancy between them and keep the best indicators to evaluate the parameters of interest. From the output variables of this pre-treatment, as many models as variables were created to explain each variable by the other ones. Then, those models were evaluated with statistical tools, and the relationships that were observed with these models were compared to the two-by-two relationships described in the literature. After evaluating the modeling part, models were used to create a virtual population of animals to study the relationships between the modeled variables with more accuracy. To compare the nutritional and sensory quality of the meat, a set of importance weights was proposed to aggregate the variables evaluating those qualities and create a synthetic index for each parameter. The correlation between both of those indexes was calculated to study their relation. Finally, the 30 best and worst profiles (based on an objective of balance between nutritional and organoleptic quality) were studied to compare their main trait differences. 

### 2.1. Data

#### 2.1.1. Animals 

Data were collected in a research project named “Lipivimus” (ANR-06-PNRA-018) investigating the effects of several lipids sources in the animal feeding on the nutritional and organoleptic qualities of the meat. Experimental procedures and animal-holding facilities respected French animal protection legislation, including the licensing of experimenters. They were controlled and approved by the French Veterinary Services (the abattoir and cattle experimental facilities license numbers were #63 345 01 and #63 345 17, respectively). A total of 30 Blonde d’Aquitaine heifers, which were homogeneous in terms of initial age (28 ± 1.3 months), live weight (540 ± 13 kg), and body condition score (2.2 ± 0.05) were raised during a finishing period of 100 days. Animals were assigned at random to one of the four rations that were i) isoenergetic on a net energy basis at a level of 1.71 Mcal of net energy/kg dry matter (DM) and ii) isonitrogenous on a metabolizable protein basis expressed in Protein Digestible in the small Intestine (PDI) units (defined by the French National Institute for Agricultural Research, or INRA, tables [6]). A lipid treatment (40 g/kg DM) was added to the basal diet made of molasses straw (30%) and concentrate (70%). The control group (*n* = 8) received the basal diet without any lipid source supplements, the flaxseed group (*n* = 8) received the basal diet added with extruded flaxseed, the rapeseed group (*n* = 6) received a mixture composed of extruded flaxseed (1/3) and rapeseed (2/3), and the palmitostearate group (*n* = 8) received palmitostearate. Two of the eight initial heifers of the rapeseed diet group are missing in the database: one for health problems was removed from the group, and the other one had not been tasted by the panel of degustation; thus, numerous data were missing for this animal. Average daily gain (ADG) was calculated based on the difference between the end and the beginning weights of the fattening period. The dry matter intake was measured daily for each pen to calculate the feed conversion ration per pen (FCR), based on a mean ADG of the animals of each pen. 

#### 2.1.2. Animal Slaughtering Process

Animals were slaughtered at the slaughterhouse of the Société Vitréenne d’Abattage (SVA) in Vitré, France, at the mean live weight of 693 kg (± 10 kg), with a body fat score of three (± 0.1) on a scale varying from one (very lean) to five (very fat). Overall, average daily gain was 1528 g/day (± 53 g/day) for the 100 day-finishing period with no significant variations between diets. Slaughtering was performed in compliance with French welfare regulations. The carcasses were not electrically stimulated, and they were chilled and stored at 4 °C until 24 h post mortem. Ultimate pH was recorded between the sixth and seventh rib using a pH meter equipped with a glass electrode at 24 h post mortem. Cold carcass weight, fat score, and conformation were measured at slaughter based on the EUROP grading system [7]. The fat, muscle, and bone proportions were estimated from the regression equations base on the dissection of the sixth rib [8]. Various fat tissues (kidney fat, pelvic fat, trimming fat) were also weighed. Professional experts from the Institut de l’Elevage scored each carcass for the intermuscular and intramuscular fat developments of the *Longissimus thoracis* muscle (LT) on two scales varying either from zero (absence of intermuscular fat) to five (strong intermuscular fat) based on an Institut de l’Elevage grading system, or from three (no intramuscular fat visible) to 12 (high intramuscular fat) [9], respectively. Meat color was monitored at the slaughterhouse on the LT muscle, at the sixth rib level twice, with a visual evaluation based on an Institut de l’Elevage grade varying from 1 (meat very light) to 4 (meat very dark) and using a portable colorimeter (CR400 MINOLTA) and D_65_ as the illuminant, because it closely approximates daylight [10]. Color coordinates were calculated in the CIELAB system [11]: *L** (lightness), *a** (redness) and *b** (yellowness). The chroma (as *C** = (*a**^2^ + *b**^2^)^1/2^) and hue angle (*h** = arctan(*b**/*a**)) were also calculated. Measurements were taken at three locations on each steak and averaged.

#### 2.1.3. Nutritional Quality Measurement

Samples of LT muscles (100−120 g) were collected one day post mortem at the level of the 9th and 10th ribs, cut into small pieces, frozen in liquid N_2_, and stored at −80 °C. Just before analysis, the frozen samples were mixed in liquid N_2_ in an analytical mill (modelM-20, IKA-Werke, Stouten, Germany) to obtain fine powder. Muscle lipids and fatty acids were extracted and quantified as previously described by Habeanu et al. [12]. Briefly, total lipids were extracted according to the method of Folch et al. [13] by mixing the LT muscle powder with a 2/1 chloroform/methanol mixture (*v/v*) and quantified by gravimetry. Fatty acid extraction and transmethylation into fatty acid methyl esters (FAME) were subsequently performed according the methods of Bauchart et al. [14]. Fatty acid methyl ester analysis was performed with gas liquid chromatography using a Peri 2100-chromatography system (Perichrom Society, Saulx-les-Chartreux, France) fitted with a CP-Sil 88 glass capillary column (Varian, Palo Alto, CA, USA; length = 100 m; diameter = 0.25 mm). The carrier gas was H_2_, and the oven and flame ionization detector temperatures described by Scislowski et al. [15] were used. Total fatty acid (FA) was quantified using C19:0 as an internal standard. The identification of each individual FAME and the calculation of the response coefficients for each individual FAME were performed using the quantitative mix C4–C24 Fame (Supelco, Bellafonte, PA, USA).

#### 2.1.4. Organoleptic Quality Measurement

At 48 h post mortem, the *Longissimus thoracis* muscle samples that were used for organoleptic evaluation were removed from carcasses (7^e^ and 8^e^ rib), placed in sealed plastic bags under vacuum, and kept at 4 °C without light for aging for 10 days. Then, each sample was frozen and stored at −20 °C awaiting organoleptic evaluation. Samples were thawed, without stacking or overlapping, for 24 h before cooking and organoleptic assessment. Then, they were cut into mini-roasts (50-g cubes, 4 cm thick). Each mini-roast was baked (thermolyne muffle furnace, Model 6000, Bioblock Scientific, Illkirch Graffenstaden, France) without dressing. There were heated to 310 °C for seven minutes, and cut into four bites that were immediately served in another porcelain plate to 12 trained panelists. The panelists rated the sample on a 10-cm unstructured line scale (from 0 to 100) measured in mm for the following texture attributes: global tenderness defined as the ease of chewing the sample between teeth: from extremely tough (0) to extremely tender (100), juiciness defined as the amount of moisture released in the mouth: from extremely dry (0) to extremely juicy (100), and global intensity of flavor from low intensity (0) to very high intensity (100). Height descriptors of flavor were also used, from low intensity (0) to very high intensity (100): sweet flavor, acid flavor, bitter flavor, fatty flavor, metallic flavor, rancid flavor, fish flavor, and blood flavor. The sessions were carried out in a room equipped with individual booths under artificial red light to reduce the influence of the appearance of the samples. At each session, a monadic presentation of a maximum of 12 samples was done, with each sample being selected in random order.

### 2.2. Modelization

#### 2.2.1. Pre-Treatment of the Variables

This experiment resulted in the collection of 62 variables (described in Appendix A), which were centered and reduced by the standard deviation. The variables were classified into one of the four parameters of interest (i.e., animal performances, carcass properties, nutritional quality, and organoleptic quality of meat). Then, a clustering of the variables was performed with the variables evaluating each parameter to reduce the redundancy among the variables and only keep a small set of indicators to evaluate those parameters. Clusters of correlated variables were formed with this statistical tool [16], and a criteria was calculated to evaluate the cohesiveness of those clusters (the closer the criteria was to 100%, the more correlated the variables that were gathered into each clusters were to each other). Then, the authors decided to resume the clusters by either a linear combination of the variables of the cluster (first component of the principal component analysis (PCA) on the variables of the cluster) or by one of the variables from the cluster. These choices are assumed to be subjective, but were based on the literature and the main indicators that were used to describe the carcass traits [17], the nutritional quality [18,19,20], and the organoleptic quality [21].

#### 2.2.2. Variable Relations Modeling 

To study the relations between the four parameters of interest, each variable was modeled with the set of all the variables that were evaluating another parameter of interest other than the one that the variable was to model. To clear the notation, let’s define M_var_ to denote the model built to explain the dependent variable var.

To determine the most relevant model as possible, several types of regression models (linear model (lm), random forest (rf), sliced inverse regression (sir), ridge regression, and partial least square regression (plsr)) were compared using modvarsel R package [22]. The underlying methodology used in this package is entirely computational, generating a training sample several times (*n* = 50), to build models that were then evaluated with a mean square error (MSE) criteria calculated on the corresponding test dataset sample. This package also provides a tool to measure the importance of each independent variable in the corresponding regression model, and then select the best ones to build the most accurate model. The quality of the model was evaluated with the adjusted coefficient of determination (adjusted coefficient of determination (adjR^2^) is a *R*^2^ penalized by the number of independent variables into the model) calculated twice: first on the training data, which were used to build the model, and then a second time with a method developed by Harrell et al. [23] based on bootstrap resampling (B = 100) to evaluate without overfitting the accuracy of the model. The combination of both methods enables estimating model overfitting. 

#### 2.2.3. Study of the Relationships Modeled

Two strategies were set up to analyze the relationships between the dependent variable and the independent variables. As mixed parametric and non-parametric models complicate the quantification of the importance of each independent variable in the prediction models, a common method was used to evaluate this importance, mobilizing the decomposition of the variance [24] in the sensitivity R package [25] by calculating a sensitivity index (Si ∈ [0, 1]). A sensitivity index was assigned to each indicator into the model: the most influential ones were associated with a value close to one and the less influential was assigned a value close to zero. The second strategy to analyze the model behavior was to visualize the variations of the model predictions where all the (centered and reduced) independent variables were set to their mean zero, except for one that was taking a range of values in 0 ± 2σ (i.e., [−2, 2] as σ = 1). This procedure was repeated as many times as the number of independent variables of the model.

### 2.3. Trade-Offs Methodology

#### 2.3.1. Aggregation of the Variable Evaluating the Parameters of Interest

Several variables were used to evaluate each parameter of interest. Therefore, to compare the nutritional and organoleptic quality, a synthetic index was created for both. Those indexes were calculated by a linear combination of the variables evaluating the parameters of interest weighted by their importance. The importance weights (Table 1) were proposed by a set of experts from the INRA (French National Institute for Agricultural Research) and Institut de l’Elevage (French Livestock Institute) based on their knowledge and the literature. For organoleptic quality, the weights were mainly based on the Meat Quality 4 (MQ4) [21] but for the nutritional quality, even if there are some considerations about the studied indicators in the literature, no objective equation has been created so far [18,19,20].

#### 2.3.2. Generation of Virtual Animals

Facing a low number of animals, the models of prediction were used to create virtual animals (*n* = 500). The aim of this approach was to study the interactions between the parameters of interest based on the models and not the real animals (which have only been used to create the models). However, it was assumed that the virtual animals were built to be realistic (i.e., those animals could be real), even if the bias of the models has to be taken into account.

The (simplified) algorithm to generate virtual animals is given in Appendix B. First, one of the parameters of interest was chosen to set the value of its variables. Each variable could vary from −2 to 2 (0 ± 2σ) with a step of 0.1. To prevent unrealistic combinations of the variables from that chosen parameter of interest, simple linear regressions were computed between these variables. A confidence interval at 95% based on those regression were decided if the combination were realistic or not. Then, all the variables that were not in the chosen parameter of interest were estimated through the prediction models. Since the models were dependent all together and the estimation of the variables made successively, all the estimation were initially set to 0. The estimations of all the variables were made several times (*n* = 10), with an order of the estimation randomly assigned, to converge to a stable state.

#### 2.3.3. Statistical Analysis

After their generation, the virtual animals were aggregated following the procedure described in Section 2.3.1. The correlation between the two synthetic indexes was computed to evaluate the link between the nutritional and the organoleptic qualities.

The trade-off method consisted of picking the best animal based on its performances regarding both of the qualities studied. To determine the best animals, a targeted point was set, and the 30 closest animals to this point were selected. The targeted point was set at the coordinate (NQ = 2, OQ = 2) corresponding to two standard deviations of the synthetic indexes (which means that only a few animals would reach such performances). The same procedure was applied to select the 30 worst animals close to the opposite targeted point (NQ = −2, OQ = −2).

Then, the best and worst selected virtual animals were analyzed. The distribution of their traits was compared by using the Wilcoxon test (robust non-parametric test which does not need to verify the normality assumption) on each variable.

## 3. Results 

### 3.1. Pre-Treatment of the Variables

The selection of the relevant variables evaluating the parameters of interest are explained in this section. The cluster affectation of each variable’s information and the correlation with the linear combination of all the variables of the cluster are given in Appendix A. A clustering of variables was performed for each parameter of interest, except for Animal Performances (APs) since the (only) three variables evaluating this parameter were uncorrelated. 

For Carcass Properties (CP), six clusters were created, which had good cohesiveness (63.7%). Intermuscular fat score and carcass fat development were associated in the same cluster, whereas the amount of the different fats (such as abdominal, fifth quarter, etc.) were all associated in another cluster. Thus, intermuscular fat appears to be only weakly linked to removed fat. This distribution of fat variables is consistent with results of Yang et al. [26], who indicated that intermuscular fat is independent from the other fat deposits. Therefore, two linear combinations were chosen to represent these two clusters (Table 2). Logically, the color variables of *a**, *b**, and *C** were positively linked, confirming the correlations already established by Mancini and Hunt [27]. The clustering of *h** and *L** color variables with fat proportion might be explained by the impact of fat on muscle color perception. Two linear combinations were also computed to represent both of these color aspects. Let us also mention that cluster CP1 was gathering carcass weight and bone proportion (logically negatively correlated) and pH. Since pH is an important indicator for the meat quality, it was proposed to separate the pH from the two other variables, which were informing more on the carcass weight. Finally, to summarize this cluster, pH and the carcass weight variables were kept as output variables of the pre-treatment operation. 

The Nutritional Quality (NQ) of the meat was evaluated by 28 variables: the lipid content and 27 fatty acid measures or the ratio/sum of them. Therefore, the decision was made to withdraw the lipid content from the analysis (because the information of this indicator is also important to compare with intramuscular fat, for example), and perform the clustering of variables only on the fatty acid variables. The cohesiveness of the six clusters was higher than that for CP (around 72%). Two clusters were not explicitly clear. The NQ3 cluster regrouped variables describing the opposition between *n*-3 and *n*-6 fatty acids. The presence of SFA and C16:0 in this cluster could be explained by a link between the low amount of *n*-3 in proportion when SFA (mainly C16:0) is high. For this cluster, it was proposed to take the *n*-6/*n*-3 ratio, which is a strong nutritional indicator, into the ANSES recommendations [28]. The NQ4 cluster was also not clear, but on the whole, it appeared to be describing the opposition between MUFA and PUFA. For this cluster, the linear combination of the variable was taken and renamed as the ratio PUFA/MUFA.

Organoleptic Quality (OQ) is usually described in the literature with three main indicators: tenderness, juiciness, and flavor. The sensory variables available in this experiment gave a larger analysis of the flavor aspect. Therefore, the clustering of variables was performed only with these variables. Five clusters were formed with a cohesiveness equal to 63.6%. The synthetic index that was chosen was strongly correlated with the input variables, meaning that a small amount of information was lost in the dimension reduction step. 

### 3.2. Choice and Quality of the Prediction Models

A synthesis of the 24 models’ quality is represented in Figure 1. Random forest was chosen most of the time (18 green intervals) before ridge regression (five red intervals) and the linear regression (one blue interval). Regarding the accuracy of the models, the adjusted *R*^2^ (adjR^2^) values were higher than 0.8 in most of the prediction models. However, some models were characterized by low or very low adjR^2^ values, especially M_ADG_, M_pH_, M_CLA_, and M_juiciness_, which denote unreliable models. 

The correction of the adjusted *R*^2^ (cor_adjR^2^) enabled taking the overfitting of the models into account, and thus estimating the true accuracy of the model on unknown data. For example, the cor_adjR^2^ of the M_FCR_ model was only around 0.33 when the adjR^2^ was higher than 0.9. Similarly, the M*_a_*_**b***C**_ and M_long FA_ appeared to be less accurate when considering their cor_adjR^2^ (0.35 and 0.38, respectively) compared to their adjR^2^ (0.95 and 0.90, respectively). It is interesting to notice that the models with high overfitting were all random forest models. 

### 3.3. Examination of Models Behavior

Some of the 24 multivariate regression models that were built were analyzed in this section. The aim of this analysis was to show the main relations between the variables to compare and discuss them further with the current knowledge (Section 4.2.). All the relations are summarized in Figure 2 by the sensitivity index (Si) of every dependent variable into each model. Only the selected variables with the modvarsel method were used to build the model and thus have a corresponding square on the row of the model. It would be too long to describe all the results in Figure 2 one by one, but to take an example, the M_FCR_ is mainly influenced by the pH (Si = 0.20), the conformation (Si = 0.20), and the CLA (Si = 0.22). The rest of the independent variables seem to be not strongly linked to the feed efficiency.

The second method that was used to describe the models’ behavior gave complementary information to the sensitivity indexes. The M_fat proportion_ seemed to be mainly influenced by the lipid content (Si = 0.90), but the graphical approach also showed that the ADG significantly influenced the model predictions (Figure 2a), even though the sensitivity index was low (Si = 0.05). As observed in Figure 2a, a low growth rate was associated with a high portion of fat and a high amount of intramuscular and intermuscular fat deposits. This example highlighted a contradiction between the two methods, which will be discussed further (Section 4.1.). 

The graphical approach was also a tool to describe the shape of the variations of the prediction. M_PUFA/MUFA_ was quite exclusively influenced by fat proportion (Si = 0.86), even if nine other variables were selected as predictors in model construction. However, it was only thanks to the graphical approach that the relation appeared to be clearly negative. Combining the previous information, it was possible to conclude that a thin carcass was characterized by a higher portion of PUFA (relatively to MUFA) than fatter carcasses. Nevertheless, this relation seemed to not be linear in the model (Figure 2b). 

Curious variations were sometimes observed. The M*_n_*_-6/*n*-3_ was built with seven independent variables, including the FCR (Si = 0.20) and the fat proportion (Si = 0.42), which were the most influential. The *n*-6/*n*-3 prediction was low for the intermediate values of FCR and fat proportions, and higher when the variables were getting close to the edges of the values tested (Figure 2c). 

M_Tenderness_ appeared to be mainly linked to the amount of fat (Si = 0.55) and the weight of the animal (Si = 0.15) or of its carcass (Si = 0.19). The sum of all the independent variables was not exactly equal to one. This observation was almost always true for all the model analyses, and will therefore be discussed further (Section 4.1.). The influence of fat development was not very clear in Figure 2d. The highest tenderness seems to be reached for intermediate fat development. The tenderness seemed to be also very low when the animals were very fat. 

Although some of the models were difficult to interpret with biological certitude, there were models where the conclusions were simpler. For instance, the M_rancid and fish flavors_ value was mostly affected by the PUFA/MUFA (Si = 0.81). In Figure 2e, it is clearly shown that after reaching a certain proportion of PUFA (relatively to MUFA), the level of unwanted flavors dramatically increased. 

Likewise, the M_flavor intensity_ was highly impacted by fat proportion (Si = 0.48) with a high intensity when the measure was high (Figure 2f). Therefore, the flavor intensity was easily linked to the intermuscular and the intramuscular fat, which partly composed the cluster and thus the linear combination calculated from it. 

### 3.4. Global Relation between Nutritional and Organoleptic Quality

The Organoleptic Quality (OQ) and the Nutritional Quality (NQ) indexes of the virtual animals (points) are displayed in Figure 3, and overlapped by the 30 real animals (triangles) used to build the models. A slight negative correlation was observed between NQ and OQ (*r* = −0.17) for the indexes calculated with the virtual animals. The same correlation with the real animals was a little bit higher (*r* = −0.25), but stayed very low, indicating that these two parameters are only weakly linked. The correlation test indicates that the correlation between the synthetic indexes calculated with the virtual animals was significantly different from zero. This test does not show the strength of the correlation and is highly influenced by the number of observations, as it can be observed with a higher p value for the real animal, even though the correlation seemed to be stronger.

### 3.5. Comparison between the Best and the Worst Profile

Among the virtual animals (Figure 4), the traits of the 30 best (blue points) and worst (red points) animals were compared in Figure 5. Overlapped on the boxplots, three real animals for each category (best and worst) were also selected (black triangles) in Figure 4 and compared with virtual ones in Figure 5 (blue and red triangles). Since they were selected for their nutritional and organoleptic quality, the best animals naturally have better traits regarding these qualities. 

Regarding the organoleptic quality (Figure 4d), the best animals had a better tenderness and flavor intensity than the worst animals. Even though the models were useless for the juiciness (predicting only the mean juiciness, because the model does not fit the variation of the juiciness at all), the best real animals (triangles) seemed to have a better juiciness than the worst real animals selected. On the abnormal flavor side, the best animals seemed to be more bitter than the worst animals, but the fatty/metal taste was stronger for the worst animals. No significant difference was reported for the rancid/fish and blood/acid flavor between the two extremes. 

Regarding nutritional quality, the best animals had lower lipid content (Figure 5c). Logically, the C16:0 proportion was lower, meaning probably that the SFA content was also lower. In addition, PUFA/MUFA was higher, which also seems logical when the SFA is low. Among the higher proportion of PUFA, the *n*-3 fatty acid seemed to have a higher proportion in comparison to the *n*-6. Associated with higher PUFA proportions, *trans* fatty acids were also higher. No significant difference between CLA proportions was observed, but the proportion of long fatty acids was higher for the best animals.

Although not selected for this parameter of interest, the worst animals seemed to have fatter carcasses that were heavier and had conformations (Figure 5b). Those fatter carcasses seemed to have an impact on the luminescence (*L**) of the meat, but not on the *a***b***C** indicator, where no difference was observed. The pH of the virtual animals seemed higher for the best virtual animals, even though this result was in contradiction with the six real animals selected.

Related to these observations on the carcass traits, the slaughter weight of the best animals was lighter. To produce fatter carcasses, it seems that the worst animals also had a lower feed efficiency (Figure 5a). No difference between the average daily gain was observed even if best real animals tended to be better.

## 4. Discussion

### 4.1. Modeling and Analytic Choices

All the approaches were based on a reliable modeling of the relations between the variables evaluating the parameters of interest. Some of the models were clearly not reliable. The worst case was that of the juiciness, which had very low adjusted *R*^2^ values, and seemed not able to predict anything other than the juiciness mean (as can be seen in Figure 5d).

The results coming out the modvarsel package will also be discussed, especially regarding the type of model selection. Among the five considered regression models in the modvarsel R package, two model types have never been selected: plsr and sir (see Figure 1). This could be explained by a lack of (linear) structure regarding the variables and the relatively low number of observations (animals) compared to the relatively high number of independent variable candidates. Thus, the dimension reduction techniques were affected and not better than the other competitive fully parametric or non-parametric regression models. Linear regression was selected only once. In contrast, random forest (a purely non-parametric approach) gave a good accuracy in this kind of situation, which explained the high rate of random forest model selected by modvarsel.

Linear regression was selected to predict the carcass weight, which was strongly correlated with the animal’s body weight before slaughter. Curiously, the slaughter weight was not fitted by a linear regression model, in which carcass weight would be the main independent variable in return. Indeed, it appears that a higher number of variables was selected by modvarsel to fit the slaughter weight (carcass weight, fat development, C16:0/C18:0 ratio, and PUFA/MUFA ratio) than to fit the carcass weight (slaughter weight and ADG). Some slight correlations between the four variables were observed, such as between carcass weight and fat development (*r* = 0.67, *p* value < 0.001) for instance, which might explain the use of penalized regression to build a more stable model, as aimed by the modvarsel algorithm. The low number of observations (*n* = 30) with influential points could also be the source of instability (high variance of the parameters’ estimation) in the linear model fitting slaughter weight.

As it was noticed in the results (Section 3.2.), all 18 random forest models had high overfitting. This is related to the non-parametric nature of the algorithm, which is known to produce high overfitting when the number of observations is low compared to the number of independent variable [29]. For instance, it was interesting to notice that the M*_n_*_-6/*n*-3_ variations were characterized by a kind of “wave”, where the ratio was favorable for intermediate values of FCR and fat proportion. These relations did not have a clear biological explanation. Nevertheless, increasing the size of the training dataset might prevent the overfitting problem, with the robustness of the non-parametric models being based on the law of large numbers.

As noticed in Section 3.3, when analyzing the models, contradictions appeared between the two methods (i.e., the sensitivity indexes (Si) and the graphical approach). For instance, regarding the M_fat proportion_ value, the ADG had a low sensitivity index, but was responsible for a significant part of the variations with the graphical method. In general, those contradictions appeared when the number of independent variables into the model was important, and the accuracy of the model was low. When both of these conditions were met, it could lead to difficulties regarding obtaining a reliable sensitivity index. Furthermore, the graphical approach was based on the variation of only one independent variable, and the other one was set to the mean (i.e., 0). Therefore, interaction phenomena were factored out of the analysis, which could also explain the opposition between both of the methods’ results.

### 4.2. Outcome from the Holistic Approach

First, this original approach has shown that there is not a strong antagonism (*r* = −0.17) between nutritional and organoleptic quality. Moreover, the best profiles selected had very satisfying meat quality regarding the quality based on the initial population of 30 animals. This observation consolidated a previous work [2], where Ellies-Oury et al. indicated that these two parameters were orthogonal to each other and thus linearly independent. These results were obtained in a population of heterogeneous animals, which came from three different breeds (Angus, Limousine, Blonde d’Aquitaine), whereas in the present work, the results were obtained on a homogenous population of Blonde d’Aquitaine females. The lack of a strong linkage between the two parameters of interest (NQ and OQ) means that a trade-off is not fully necessary to maximize NQ and OQ simultaneously.

Going deeper in the best profiles analysis, the thin carcasses with a low proportion of fat seemed to drive to high PUFA and w3 proportions in the meat, as well as a higher intensity of flavor and tender meat. In contrast, the fattest carcasses seemed to produce meat with high lipids content that lacked PUFA and had an imbalanced *n*-6/*n*-3 ratio. The models explain these results. In Section 3.2, a negative relation was shown between the fat proportion and the PUFA/MUFA ratio. This result support those from Warren et al. [30] indicating that muscles with higher lipid content have a lower proportion of PUFA compared to MUFA and SFA. These could be explained by a higher *de novo* synthesis of fatty acid, which was mostly saturated and monounsaturated. In leaner meat, the triglycerides/phospholipids ratio was higher, and it is well known that PUFA mainly esterified on phospholipids [31]. 

In the literature, studies have shown a positive relation between fat and tenderness [32]. However, the model built in this study was different. As noticed in Figure 2d, the tenderness optimum is reached when the fat development is between −1 and 0 (which means that the animal is close to the mean or lower by one standard deviation), and the worst tenderness predicted was obtained with an upper fat development that was one standard deviation over the mean. This result was also observed in the best and worst profiles. The contradiction between this model and the literature cannot be explained by a low model quality, because the accuracy of the M_tenderness_ is quite satisfying (adjR^2^ = 0.85 and cor_adjR^2^ = 0.60). However, the model was fit on a specific population of 30 Blonde d’Aquitaine heifers. This breed is well known to produce very lean meat in comparison to the other suckling breeds [33], which could explain the different relationship between this fat indicator and the tenderness of the meat.

As previously shown, the best animals had higher PUFA/MUFA in the meat. However, high PUFA content can also lead to the risk of unwanted flavor (e.g., the fish flavor indicated in Figure 2e). This effect seems to be related to PUFA oxidation [5]. Nevertheless, the higher level of PUFA in the best profile meat did not cause this kind of abnormal flavor. 

Finally, these results confirmed that meat with higher intramuscular and intermuscular fat increases the flavor intensity [4], indicating that intramuscular fat is a precursor of many aromatic compounds formed, notably, during cooking processes. 

### 4.3. Limits and Perspectives of the Trade-Off Method

Generating virtual observations is unusual in the animal research field, and might raise interrogations and doubts regarding the results for the reader. However, there are no statistical aberrations, because this study did not try to show the impact of one factor (the diet effect for instance) on the beef performances or meat qualities or the strength of relation between two variables. All the probabilistic tests that were performed in this paper (the Wilcoxon test of comparison between the best and worst profiles traits) should not be interpreted literally; they only demonstrate that with the relation modeled in this study, the best profiles are significantly different than the worst ones regarding various traits. 

Aside from this innovative approach, the trade-off method seemed to be highly sensitive to the weighting granted for each variable by the experts. For instance, tenderness (whose weighting is significant) was the highest among the best profiles (and lowest among the worst profiles). Then, the result of optimizing was clearly dependent on the expertise provided for the relative importance of each variable evaluating the parameters of interest. Therefore, the choice of variable weights has to be carried out carefully and wisely to manage the trade-offs as efficiently as possible. 

The weighting dependence can also be related to the method used to aggregate the variables, considering its numerous disadvantages. The linear combination calculated here to aggregate the variables led to compensation between them. For instance, a high proportion of PUFA would compensate for a high *trans* fatty acid proportion. No threshold was taken into account as a way to prefer a more balanced animal compared to an animal performing well on only a few indicators. More complex methods exist to resolve these problems such as outranking methods, but suffer from less clarity and are difficult to assess [34]. In addition, the weight-related techniques need variables varying in the same scale and the same direction. In the case of some of them, such as the carcass weight, the optimum value is not a maximum or a minimum weight, but rather a homogeneous weight distributed around a target value.

## 5. Conclusions

The present paper has exposed a very innovative approach to assess the trade-off between parameters of interest in the beef cattle industry. It showed that there is no antagonism between organoleptic quality and nutritional quality. The modeling approach has also been very interesting to highlight the relation between the variables and show that they are interconnected, and not only by two-by-two relations. This knowledge should be taken into account when designing new product brand or label expectations.

However, the very strong influence of the weight setting and the aggregation method on the results of the trade-off was observed. This should encourage the meat beef industry to clarify their own expectations regarding these two parameters of interest in order to set consensual indicators with associated hierarchy to measure the organoleptic and nutritional quality.

Moreover, the significance of these results is only relevant in the particular context of the data in terms of the animals’ breed, age, diets, fattening duration, muscle, etc. Thus, the extrapolation of the present results has to be done with caution.

## Figures and Tables

**Figure 1 foods-08-00197-f001:**
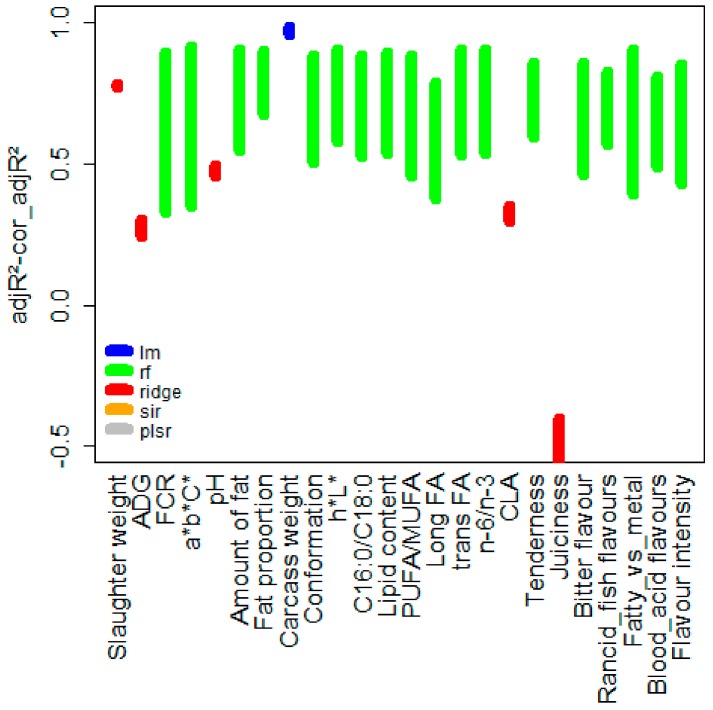
Quality of the 24 prediction models based on the adjusted coefficient of determination calculated twice, represented by vertical segments. The upper (respectively lower) limit of the interval corresponds to the adjusted *R*^2^ (calculated on the training data) (respectively corrected adjusted *R*^2^ estimate with a bootstrap approach [23]). The color indicates the model selected from the five competing regression models (lm = linear model, rf = random forest, ridge, sir = slice inverse regression, plsr = partial least square regression).

**Figure 2 foods-08-00197-f002:**
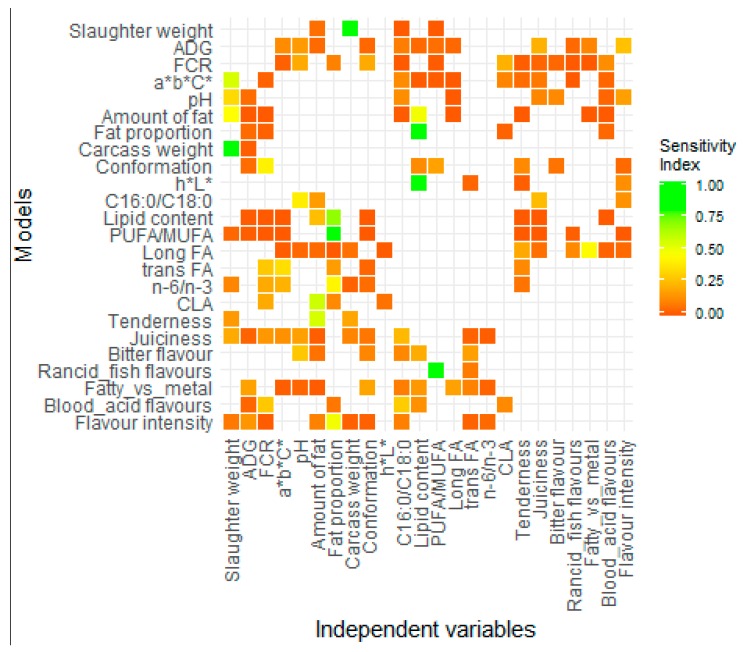
Heat map of the sensitivity index for each variable used (column) in each model (row). Green indicates values that are higher in the sensitivity index. If there is no square, the variable was not used in the corresponding model.

**Figure 3 foods-08-00197-f003:**
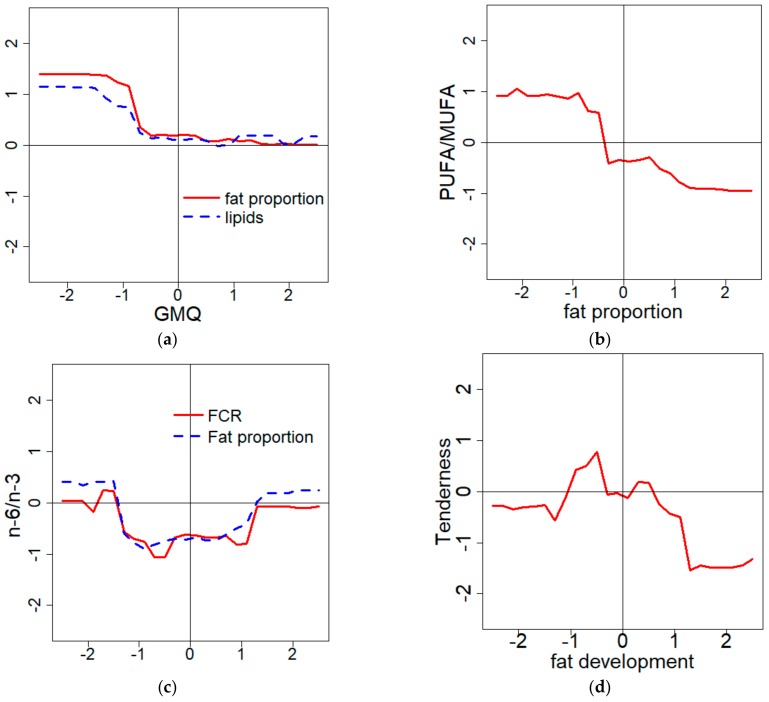
(**a**) Prediction of the fat proportion and the lipid content versus the ADG (average daily gain) variations; (**b**) Prediction of the PUFA/MUFA (polyunsaturated fatty acid/monounsaturated fatty acid)ratio versus the fat proportion variations; (**c**) Prediction of the *n*-6/*n*-3 ratio versus the FCR (feed conversion ratio) and fat proportion variations; (**d**) Prediction of the tenderness versus the fat development variation; (**e**) Prediction of the unwanted rancid-fish flavor versus the PUFA/MUFA ratio variations; (**f**) Prediction of the flavor intensity versus the ADG and the fat proportion variations.

**Figure 4 foods-08-00197-f004:**
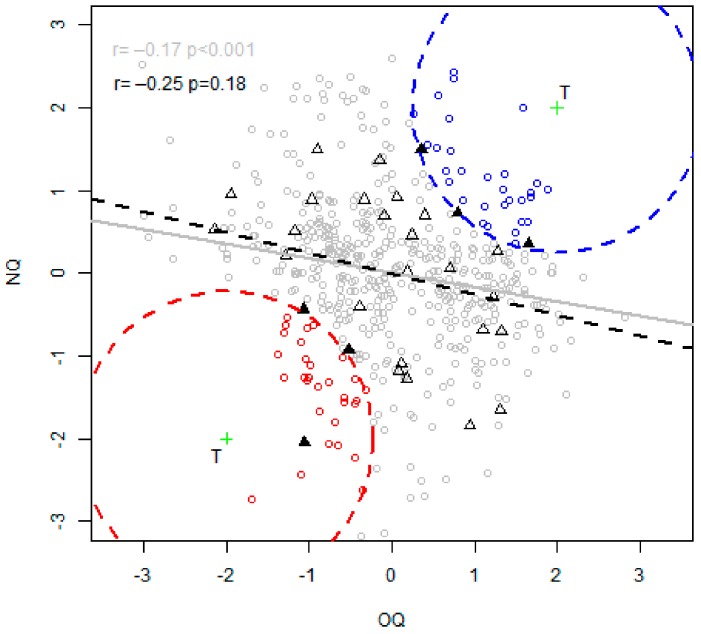
Nutritional and organoleptic indexes of the virtual (points) and real (triangles) animals. The correlation of the virtual (respectively real) animals is given in grey (respectively in black) in the top left corner. Regression (NQ~OQ) line for both virtual and real animals is added to visualize the correlation. Two targeted green crosses T were set to select the closest best (blue) and worst (red) virtual animals (and real animals filled in black for both categories).

**Figure 5 foods-08-00197-f005:**
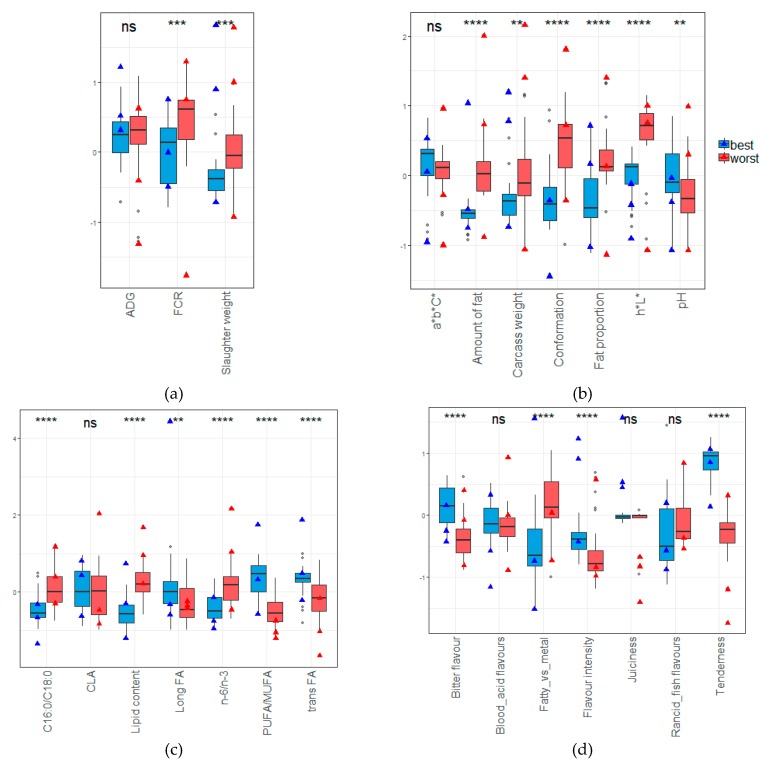
Comparison (boxplots) of the virtual animals traits between the worst (red) and the best (blue) profiles selected. The three best and worst real animals are added on the boxplots with triangles. For each variable, the result of a Wilcoxon test is provided (ns: no significant, **: *p* value < 0.01, ***: *p* value < 0.001, ****: *p* value < 0.0001). (**a**) animal performances; (**b**) carcass properties; (**c**) nutritional quality; (**d**) organoleptic quality.

**Table 1 foods-08-00197-t001:** Importance weights related to each variable evaluating the nutritional and organoleptic qualities.

Parameter of Interest NQ	Weights	Parameter of Interest OQ	Weights
lipid content	−0.15	tenderness	+0.425
long FA	+0.05	juiciness	+0.150
C16:0/C18:0 ratio	−0.15	flavor intensity	+0.175
*n*-6/*n*-3 ratio	−0.25	bitter flavor	−0.025
PUFA/MUFA ratio	+0.25	rancid and fish flavors	−0.125
CLA	0.1	fatty vs. metal	−0.050
*trans* FA	−0.05	blood and acid flavors	−0.050
Total (in absolute value)	1	Total (in absolute value)	1

CLA: conjugated linoleic acids, FA: fatty acid, MUFA: monounsaturated fatty acids, PUFA: polyunsaturated fatty acid, NQ: nutritional quality, OQ: organoleptic quality.

**Table 2 foods-08-00197-t002:** Output variables of the pre-treatment operation describing the four parameters of interest (PI).

PI	Cluster Codification	Output Variables Name/Abbreviation	Output Variables Description
Animal Performances (AP)	-	Slaughter weight	Slaughter weight
	-	ADG	Average daily gain during the finishing period
	-	FCR	Feed conversion ratio (ADG/feed intake (DM))
Carcass properties (CP)	CP1	Carcass weight	Carcass weight
		pH	Ultimate pH at 24 h post mortem
	CP2	Conformation	Carcass conformation
	CP3	h*L*	Aggregation of hue (*h**), luminosity (*L**) of carcass lean, and expert evaluation of carcass muscle color
	CP4	Fat development	Aggregation of several fat tissues (in the fifth quarter, carcass fat, etc.)
	CP5	Fat proportion	Fat percentage relative to the other components of the carcass (bone and muscle), aggregate with intermuscular and intramuscular fat score
	CP6	*a***b***C**	*a**, *b**, and chroma (*C**) of carcass lean
Nutritional Quality of meat (NQ)	-	Lipid content	Lipid content
	NQ1	Long FA	Long-chain fatty acid amount and proportion
	NQ2	C16:0/C18:0	C16:0/C18:0 ratio
	NQ3	*n*-6/*n*-3	*n*-6/*n*-3 ratio
	NQ4	PUFA/MUFA	Polyunsaturated fatty acids/Monounsaturated fatty acids ratio
	NQ5	CLA	Conjugated linoleic acids
	NQ6	Trans FA	Trans fatty acids
Organoleptic Quality (OQ)	-	Tenderness	Tenderness
	-	Juiciness	Juiciness
	OQ1	Flavor intensity	Flavor intensity
	OQ2	Bitter flavor	Bitter flavor
	OQ3	Rancid fish	Flavors rancid and fish flavors
	OQ4	Fatty vs metal	Fatty versus metallic flavors
	OQ5	Blood acid flavors	Blood acid flavors

ADG: average daily gain, FCR: feed conversion ration per pen.

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
