# Peer review of "New Approach Studying Interactions Regarding Trade-Off between Beef Performances and Meat Qualities"

_foods, 2019, doi:10.3390/foods8060197_

Round 1
Reviewer 1 Report
This work has a good scientific quality and is well designed. The results was argued well, comparing them to earlier results and contributing new ideas. However, need to revise some questions to be published. I recommend a minor revision.
Comments to authors (e.g. suggestions of changes to the text.):
Table 1 should be revised since the units of some variables are missing. Although the authors specify that they are weight, it seems to me correct to specify between parentheses (kg).
Author Response
Dear reviewer 1,
Thank you very much for your positive comments on our work. Your global comment was:
“This work has a good scientific quality and is well designed. The results was argued well, comparing them to earlier results and contributing new ideas. However, need to revise some questions to be published. I recommend a minor revision.”
However, the second reviewer was more critical on the article and we had to modify the manuscript a lot to satisfy his remarks. I hope the changes made in the new version do not change your interest in the paper. We tried to improve the English level but we were running out of time after clarifying the points of the reviewer 2.
Your comment about the Table 1:
“Table 1 should be revised since the units of some variables are missing. Although the authors specify that they are weight, it seems to me correct to specify between parentheses (kg)”
Our reply: Thank you very this remark which led us to precise the table caption. Actually, the weights are not expressed in kg in the table, they are mathematical importance weights used to aggregate the variables as a linear combination and create the two indexes studied further.
Once more, we would like to thank you for your careful reading and insightful comments.
Regards,
Alexandre Conanec, Brigitte Picard, Denis Durand, Gonzalo Cantalapiedra-Hijar, Marie Chavent, Christophe Denoyelle, Dominique Gruffat, Jérôme Normand, Jérôme Saracco and Marie-Pierre Ellies-Oury
Reviewer 2 Report
I'm sure the thesis will have quality. But the author wanted to include so much information in the article that nothing is understood. Honestly, I think the usefulness of this article is the statistical method, so I would reccomended to the author eliminate all the part that includes real animals and then it is not used, and dedicate their efforts to explain how works the used statistical tool. Because in fact, it can not be said that the results were "zootechnically novel" (much less for INRA). But to study the "usual" data in another way can shed some light on the relationships between variables, which is always interesting because it allows to synthesize the huge amount of available information and, as indicated in the article, to give indications or simple rules to the industry. So I encourage the author to throw it all away and start over, focusing only on the statistical part.

Author Response
Dear reviewer 2,
We would like to express our gratitude for your careful reading of the original manuscript and for your comments and suggestions.
Your global comment was:
“I'm sure the thesis will have quality. But the author wanted to include so much information in the article that nothing is understood. Honestly, I think the usefulness of this article is the statistical method, so I would reccomended to the author eliminate all the part that includes real animals and then it is not used, and dedicate their efforts to explain how works the used statistical tool. Because in fact, it can not be said that the results were "zootechnically novel" (much less for INRA). But to study the "usual" data in another way can shed some light on the relationships between variables, which is always interesting because it allows to synthesize the huge amount of available information and, as indicated in the article, to give indications or simple rules to the industry. So I encourage the author to throw it all away and start over, focusing only on the statistical part.”
Our reply: Thank you very much for all your comments which show us that the article was not clear enough. We followed your advice and re-wrote almost entirely this article with a new angle on the statistical tool. With these modifications, the aim of the article is focused on explaining the method analyzing the relations between beef performances and meat quality as a whole. However, we still described the real animals in this paper because they are the base of the models.
Besides the proposition you have made to clarify the article, we will now try to answer point by point the questions that you raised into the article:
1) In the introduction (lines 40-42), you contest the sentence saying that only “few studies have been carried out to tackle the problem as a whole, describing the relation between these four parameters of interest: animal performances, carcass properties, nutritional and organoleptic qualities”. Actually, we are not saying that there is no paper studying the interactions between the parameters of interest, but we are not aware of papers describing them with an original approach to synthetize those interactions as ours. If we are wrong, please feel free to give us some example.
2) Your other interrogation (lines 143, 201, 203, 218, 221) in the text was how we reached the output variables of the pre-treatment described in the Table 1. This interrogation is obviously understandable given the few details in the previous manuscript. We did so, because we thought that it was not the most original result and certainly not the most interesting point to develop but we were clearly wrong because the reader is confused. Therefore, we gave more details in the new version and we added clusters information on the table in appendix A.
3) Linked with the previous point, you also had difficulty to understand the difference between the cluster, the variable summarizing each cluster and the model built for each variable. We hope that with the explanation and the new notation of the model Mvar it will be more clear. Also, the analysis of the model seemed not clear, specially the sensitivity index part, obviously because we did not give a figure to summarize all the results. In the new manuscript, we added a heatmap which has the benefit to give all the sensitivity index (with a color scale) but also to show which independent variables are in each model.
4) You made us couple of time a remark about the low number of animal in the study (lines 70, 178). We agree and we are aware that 30 animals are not a lot. However, we have used in consequence an objective method to evaluate the models to take into account the potential influence of few points. We also indicate in our conclusion that the present results are related to a precise context (line 509-511 in the new manuscript).
5) In lines 175-176, you explained that some choices seemed to be done subjectively like the weights given by a set of expert from INRA and IDELE. Indeed, this modeling work is based on subjective choices that are assumed. As we clear it out in the new manuscript, nowadays, no equation exists to define the nutritional quality on the base of fatty acid variables. Therefore, to develop our approach we had to make a proposition which can be discussed but which was necessary.
6) At the line 231, you ask us to discuss when a coefficient of determination is robust. This is not a very easy question because there is no threshold to answer at your question. R² have to be studied in a context and we think that the answers brought in the new manuscript are the most we can do to interpret the quality of the models.
7) Line 243, you ask why we choose to develop only these “7 models”. Actually we developed them because they were useful in the discussion. However, in the new manuscript, we chose to explain them only to illustrate the validity of the model, that there are close (or not) to the already known two by two relations.
8) Line 291, you interrogate us on the low correlation (r=0.17) and the p.value associated. First, the null hypothesis considered in the test is H0: ρ = 0 versus H1: ρ 0, where ρ is the population correlation. the p.value tested the hypothesis of null correlation. A weak empirical between the indexes can be significantly different from zero (p.value < 5%, reject of H0) because of a large sample size. Hence, this significant p.value does not mean that the underlying correlation is strong. We hope that it is better written in the new manuscript (line 354-356).
9) You interrogate us why we chose to pick only 2 best and 2 worst profile to study them. You are right, this was a bad choice and we changed it in the new version. We finally decided to select 30 virtual animals which are more representative of the distribution to compare their distribution to the opposite worst virtual animals selected.
Once more, we would like to thank you for your careful reading and insightful commetns. Your comments and suggestion led to a significant improvement of the original manuscript. We hope that we succeeded in clarifying the issues raised and that the paper can be considered for acceptance.
Regards,
Alexandre Conanec, Brigitte Picard, Denis Durand, Gonzalo Cantalapiedra-Hijar, Marie Chavent, Christophe Denoyelle, Dominique Gruffat, Jérôme Normand, Jérôme Saracco and Marie-Pierre Ellies-Oury
Round 2
Reviewer 2 Report
I congratulated the authors for the work done (and I appreciate it). They have worked a lot but it was much clearer how things were done